

# Arrhythmia classification for non-experts using infinite impulse response (IIR)-filter-based machine learning and deep learning models of the electrocardiogram

Mallikarjunamallu K and  Khasim Syed

School of Computer Science and Engineering, VIT-AP University, Amaravati, Andhra Pradesh, India

## ABSTRACT

Arrhythmias are a leading cause of cardiovascular morbidity and mortality. Portable electrocardiogram (ECG) monitors have been used for decades to monitor patients with arrhythmias. These monitors provide real-time data on cardiac activity to identify irregular heartbeats. However, rhythm monitoring and wave detection, especially in the 12-lead ECG, make it difficult to interpret the ECG analysis by correlating it with the condition of the patient. Moreover, even experienced practitioners find ECG analysis challenging. All of this is due to the noise in ECG readings and the frequencies at which the noise occurs. The primary objective of this research is to remove noise and extract features from ECG signals using the proposed infinite impulse response (IIR) filter to improve ECG quality, which can be better understood by non-experts. For this purpose, this study used ECG signal data from the Massachusetts Institute of Technology Beth Israel Hospital (MIT-BIH) database. This allows the acquired data to be easily evaluated using machine learning (ML) and deep learning (DL) models and classified as rhythms. To achieve accurate results, we applied hyperparameter (HP)-tuning for ML classifiers and fine-tuning (FT) for DL models. This study also examined the categorization of arrhythmias using different filters and the changes in accuracy. As a result, when all models were evaluated, DenseNet-121 without FT achieved 99% accuracy, while FT showed better results with 99.97% accuracy.

# INTRODUCTION

The heart is a critical organ. It controls the operation of the blood circulatory organs (*Badr et al., 2022*). When these organs struggle to supply enough blood to the heart, heart disease (HD) occurs. HD kills the majority of people worldwide. HD is the top cause of death among American Indians, according to the American Heart Association (AHA) (*Sharma et al., 2023*). It is the most common cardiovascular disease in women over 65 years of age (*Tsao et al., 2022*; *NHS, 2022*). Cardiac arrhythmias are a common form of HD. Arrhythmia affects the heartbeat. It indicates a fast, slow, or irregular heartbeat. Tachycardia is a fast heartbeat.

Corresponding author
Khasim Syed,
profkhasim@gmail.com,
syed.khasim@vitap.ac.in

Atrial fibrillation is the most common type of heart arrhythmia (*NHLBI, 2022*). ECGs are one of the simplest methods to check heart rhythm and can easily and quickly identify cardiac arrhythmias. A portable Holter monitor was used to record ECGs. It can record ECG signals for 24 h (*Hopkins Medicine, 2022*). A huge number of people suffer from irregular heartbeats regularly (*Chowdhury, Poudel & Hu, 2020*), which can be dangerous. Thus, an accurate, low-cost arrhythmic heartbeat diagnosis is desirable. ECG signals, which show heart electrical activity in P, QRS, and T waveforms, have been used by many researchers to classify arrhythmia.

Time, size, and distance between waves and peaks determine heart arrhythmia. Feature extraction and beat classification help diagnose arrhythmia. Depending on the severity of the symptoms present upon diagnosis, cardiac arrhythmias are categorized as either life-threatening or non-life-threatening. Potentially fatal arrhythmias like ventricular fibrillation and tachycardia can cause cardiac arrest and rapid death. These patients require immediate medical assistance. Although non-life-threatening arrhythmias do not directly lead to heart failure, they require prompt treatment to prevent further damage. Arrhythmias may affect a patient's everyday life in various ways (*Sannino & De Pietro, 2018*). Furthermore, numerous modern medical applications have greatly elevated the significance of ECG classification for arrhythmia diagnosis. In recent years, various approaches have been developed for categorizing ECG data for arrhythmia detection. The classification accuracy of electrocardiograms (ECGs) depends on the classifier capability and ECG feature identification abilities (*Alarsan & Younes, 2019*).

An ECG has a variety of noise components, including device power interference, baseline drift (low-frequency signal variance), skin-electrode contact noise, and motion artifacts (a patient problem that happens when the patient performs an ECG either purposefully or accidentally). Arrhythmia might be misconstrued as motion artifacts from the patient's muscular action, even if the movement is safe. The frequency of ECG impulses changes with time. As a result, nonlinear noise and artifacts have an impact on ECG signals (*Kumar & Sharma, 2020*). Ambulatory electrocardiograms (AECGs) have increased significantly compared with Holter monitoring systems. AECGS ranges from 24 to 48 h in duration. AECGs range from full-scale 12-lead ECGs to small patches with narrow vectors. AECGs are used clinically for arrhythmia analysis. They are now used to classify and predict risk and to study the ST level and QT interval shape (*Xue & Yu, 2021*).

After considering all of the above observations, it can be concluded that ECGs employed for rhythm classification have built-in noises. Some of the more serious issues related to these noises are discussed below.

- It is important to have a reliable way to mark heartbeats and measure their features. Most likely, the signs of arrhythmia cannot be seen when the ECG signal is taken (*Tang et al., 2022*).
- People's ECG signals are different because their intrabeat and interbeat time amplitudes are different, as is their way of life. It is difficult to find a general framework for classifying heartbeats that can be used in a larger population (*Tripathi et al., 2022*).

- In the temporal frequency domain, the manner in which arrhythmia manifests is random. Therefore, the ECG signal study may have to be performed for a longer time. Therefore, tachycardia is more likely to be incorrectly diagnosed (*Ishaque, Khan & Krishnan, 2022*).
- Noise and other unwanted variables can mix and cover morphological patterns, making it difficult to determine the type of heartbeat (*Yeo et al., 2022*).

This study is based on a technique for removing noise present in raw ECG signals using a computer-aided diagnostic system. This can help cardiologists, especially non-experts, diagnose arrhythmias using ECG in a smart, efficient, and cost-effective manner. To achieve this, the proposed method can clearly identify five types of arrhythmias with the help of pre-filtered ECG patterns using machine learning and deep learning techniques and is based on the concept of a hybrid classification scheme.

This article is organized into sections. The first section discusses the introduction in depth. The relevant study findings are presented in the Related Work section, where they are used to predict the severity of arrhythmia, and the fundamentals of ML and DL are addressed. The Materials and Methods section discusses basic ECG knowledge, preprocessing, and the proposed IIR filter. Furthermore, information about the proposed model can be found in the Proposed Methodology section. The results and analysis section goes over the various ML and DL models. The performance discussion section compares the performance of the IIR filter with that of other types of filters. The Conclusion section explains the conclusions and any new features.

## RELATED WORK

Arrhythmia is the most common type of heart disease. Classification of arrhythmia using ECG signals is a common method. Using this method, many researchers have conducted investigations using ML and DL methods to classify rhythms. Several approaches used in the past to classify arrhythmias based on these are discussed in this section.

### Machine learning methodologies

Machine learning has become increasingly popular in recent years and it has been employed to effectively address critical challenges across multiple fields, including medicine, security, and communications (*Chen et al., 2021*). Many studies have been based on the incredible ability of machine learning to detect the existence of irregular heartbeats in ECG readings (*Luz et al., 2016*; *Irfan et al., 2022*). Several traditional classification models are being used to handle this issue. Random forest and support vector machine (RF and SVM) (*Bhattacharyya et al., 2021a*), k-nearest neighbors (KNN) (*Sinha, Tripathy & Das, 2022*), and artificial neural networks with logistic regression are the most common methods (ANN with LR) (*Sanamdikar, Hamde & Asutkar, 2020*). Different methods, like decision trees (*Mohebbanaaz & Rajani Kumari, 2022*), hidden Markov models (*Sadoughi, Shamsollahi & Fatemizadeh, 2022*), and hyperbox classifiers (*Hosseinzadeh et al., 2021*), are also used to classify arrhythmia. Classifiers such as linear discriminants (LD) (*Krasteva et al., 2015*), decision trees (*Sultan Qurraie & Ghorbani Afkhami, 2017*), and as sophisticated

as traditional neural networks (*Inan, Giovangrandi & Kovacs, 2006*; *Javadi et al., 2011*) are some of the methods available. A lot of work has also gone into finding the optimal combination of features and sometimes even making complicated signal processing methods, as well as selecting the most efficient set (dimensionality reduction) for classifying arrhythmias (*Li et al., 2017*).

Boser, Guyon, and Vapnik invented SVM in 1992 (*Vikramaditya, 2006*). Math-based and human-guided, it is a popular machine-learning algorithm. It has been proposed to solve many medical, engineering, text classification, image segmentation, and pattern recognition issues. SVMs were first used for binary classification, where they find a decision boundary (called a "hyperplane") that divides data into two classes. SVMs with different kernels work well with large data sets like the ECG signal. Only a few kernel functions can classify biomedical signals. Because it distinguishes ECG signals well, the radial basis function (RBF) kernel SVM classifier has been widely used (*Bhattacharyya et al., 2021b*).

Supervised machine learning, especially for classification tasks, is popular with the K-nearest neighbors (KNN) algorithm. However, it is a "non-parametric lazy algorithm", unlike the other methods, which fit training data differently. KNN does not train. Math groups things instead. The KNN sorts feature vectors by their closest training sample labels in the feature space. Calculate the distance (Hamming, Euclidean, Minkowski, *etc.*) between a feature vector or a new instance and all of the feature vectors in the training set to find the k-nearest neighbor. The class with the most votes is used to predict the unknown feature vector. Recent ECG classification studies have used the KNN (*Mahanya & Nithyaselvakumari, 2022*).

The random forest (RF) approach is an ensemble learning strategy that generates a large number of individual decision trees (DTs). A decision tree is a non-parametric supervised learning system that employs a majority voting mechanism to integrate predictions for classification and regression tasks. Because it only uses a portion of the features used to create them, RF can avoid deep DT overfitting. When training an RF model, the number of estimators (trees) is essential (*Mazidi, Eshghi & Raoufy, 2022*).

One of the most popular and useful teaching tools is the decision tree. Several studies have successfully classified heartbeats using decision trees (*Charfi & Kraiem, 2012*). Using wavelet transformations (*Zhang, Peng & Yu, 2010*) retrieved features from electrocardiograms, which they then used to cluster using a decision tree. An ensemble of weak learners, or base prediction models, is used to generate a prediction model in the machine learning technique known as "Gradient Boosting", which is applied to regression and classification issues. One of the most popular machine learning tools in recent years is eXtreme Gradient Boosting (XGBoost) (*Chen & Guestrin, 2016*), an improved optimization of the gradient boosting technique that incorporates various algorithmic and system improvements (*Li et al., 2019*; *Shi et al., 2019*). Latent growth mixture modeling (LGMM) is used by *Rountree-Harrison, Berkovsky & Kangas (2023)* to look at biomarkers in the heart and brain that show high-stress levels. In a recent study, *Varalakshmi & Sankaran (2022)* used the Bagging classifier to categorize arrhythmias. It works as a meta-estimator ensemble, with several copies of the core estimator. To change the training data set, a

sampling strategy is utilized. The aggregate prediction is the result of a voting mechanism that adds each estimator's estimates together.

In this way, many studies have been conducted on arrhythmia classification using ML methods and concepts.

## Deep learning methodologies

Recently, researchers have been using deep learning more and more to pull out features. Researchers have used DL models of CNN (*Ahmad et al., 2021*), CNN-LSTM (*Essa & Xie, 2021*), LSTM-AE (*Hou et al., 2019*), and BiLSTM (*Li et al., 2022*) models of neural network convolution to classify irregular heartbeats.

In reality, researchers have to do a lot of work to get features, and sometimes the features they make by hand can't accurately describe an electrocardiogram. Deep learning is much better than machine learning at dealing with big data (*Farhan & Jasim, 2022*), analyzing time series data, classifying images, *etc*. Deep learning has made a lot of progress recently that has helped make health care better. Deep learning does a great job with a lot of data, as has been shown. Deep learning can save time when it comes to extracting features and does not require a lot of related knowledge, which makes it very efficient. Researchers classify ECG data and find out how well it can be transferred by looking at Alex Net, VGG-16, ResNet-50, and the Inception CNN network typologies. In most cases, CNN's network topologies work better than those of other networks, but it takes other networks longer to process the same data, which is not practical. Aside from that, though, the ECG data is not well balanced because there aren't many negative samples. So, data that is not balanced could change the final classification (*Ali, Kareem & Mohammed, 2022*). Deep learning (DL) uses ''training data'' to learn, predict, improve decisions, or find complex patterns. CNNs are more practical than traditional learning methods because you can usually improve their accuracy by increasing the network or training dataset. Decision trees and support vector machines (SVMs) require a lot of data and human input to be generalizable, making them unsuitable for many modern applications.

Currently, deep learning (DL) architectures like Alex Net, VGG16, and ResNet-50 have been proposed to improve learning task accuracy (*Ebrahimi et al., 2020*). An eight-layer convolutional neural network called AlexNet is a trained network stored in ImageNet. The trained network can classify photos into 1,000 categories, including animals, keyboards, mice, and pencils. Thus, the network can represent many images using many features. Pictures sent to the network can only be 227 by 227. AlexNet's best feature is image direct-to-classification. Convolution layers automatically find image edges, and fully connected layers can learn them. More convolutional layers may simplify visual patterns (*Eltrass, Tayel & Ammar, 2021*).

The VGG-16 model is utilized for image classification. The system shows a high level of accuracy, correctly categorizing 1,000 photographs into 1,000 distinct categories with a success rate of 92.7 percent. This approach to categorizing images supports the process of transfer learning. This model, which is a convolutional neural network consisting of 16 layers of fully connected layers, was trained on the MIT-BIH dataset to accurately classify

electrocardiogram (ECG) rhythms. Convolutional, fully connected, and pooling layers are commonly utilized in network models (*Jun et al., 2018*).

ResNet-50 has an advanced design that is very accurate and works well with other networks. The framework adds a module for quick connections to learn the residue and stay away from deep network problems. The direct sharing of data across the network makes it easy to find high-level features that are conventional (*Zhao et al., 2022*).

The DenseNet-121 architecture has also been used to classify electrocardiogram (ECG) signals (*Cai et al., 2022*). Each layer in a typical convolutional network is solely connected to the one below it. DenseNets, on the other hand, has direct connections between each layer and the layers that follow it. Each succeeding layer will now contain the feature maps from the previous layers. To achieve the same level of performance, fewer parameters are required than in traditional convolutional networks. DenseNets support in the strengthening of convolutional networks. We have seen how advanced AlexNet (*Eltrass, Tayel & Ammar, 2021*), which has eight layers, VGG (*Jun et al., 2018*), which has sixteen, and ResNets (*Zhao et al., 2022*), which has more than 100 and even 1000 levels. DenseNets make layer interconnections easier than in other systems. As a component of DenseNet, the Dense Block serves an important role in improving information flow across layers. Batch normalization (BN), rectified linear activation unit (ReLu), and convolution (conv) are the key elements.

Similarly, several studies have been conducted on arrhythmia classification using DL methods and concepts. What we finally noticed was that all these findings persisted as long as heart disease existed. The Table 1 below provides a summary of the subjects covered so far.

## MATERIALS AND METHODS

This section discusses the database that was used, the preprocessing step, how the heat beats were found, how the classes were matched and given names, how the classes were put together, and the proposed filter design.

### Arrhythmia database

Typically, electrocardiogram (ECG or EKG) data, which is a time series of the heart's electrical activity, is included in the MIT-BIH Arrhythmia Database. The data structure consists of numerical values that represent the voltage as it changes over time, rather than images. To train and test the model suggested by *Ullah et al. (2022)*, they used a stratified 5-fold evaluation strategy on 97,720 and 141,404 transformation beat images taken from the MITBIH and St. Petersburg Institute of Cardiological Technics (INCART) datasets, which are both imbalanced class datasets. N (normal), V(ventricular ectopic), S (supraventricular ectopic), and F (fusion) are the four classifications of the data according to the Association for the Advancement of Medical Instrumentation® (AAMI).

The Mit-Bih database is used with numerical values of time series data in this article; because convolutional neural networks (CNNs) are so useful for processing images, extracting features, and classifying time series data, there is no need to convert the database

**Table 1  The study that is related to the classification of ECG signals.**

| Researcher and year | Data sources | Method | Extracted features | Approaches to classification | Accuracy |
|---|---|---|---|---|---|
| *Keskes et al. (2022)* | PhysioNet CinC 2011 | Multi-objective optimization (MOO) method | Statistical Frequency domain and Time domain features | KMeansSMOTE and SMOTETomek | KMeansSMOTE: 0.918 SMOTETomek: 0.8936 |
| *Bhattacharyya et al. (2021a)* | MIT-BIH arrhythmia dataset | Ensemble of RF and SVM | Spectral Features, Statistical Features and Temporal Features | Random Forest (RF) SVM Ensemble RF + SVM | 98.21 |
| *Sinha, Tripathy & Das (2022)* | MIT-BIH arrhythmia dataset | Empirical Mode Decomposition | Multilayer Similarity Coefficients, Time Frequency Variation and Phase Synchrony Features | DNN:Deep Neural Network LS-SVM:Linear Square SVM | DNN:99.05 LS-SVM:98.82 |
| *Sanamdikar, Hamde & Asutkar (2020)* | MIT-BIH arrhythmia dataset | General sparsed neural network (GSNN) | Time Domain Features Frequency Domain Features High Level Features | General sparsed neural network (GSNN) | 0.98 |
| *Mohebbanaaz & Rajani Kumari (2022)* | MIT-BIH arrhythmia database | Adaptive boosted optimized DT classifier | Temporal Features Morphological Features | Optimized DTAdaptive boosted optimized DT | Optimized DT:97.30 Adaptive boosted optimized DT:98.77 |
| *Sadoughi, Shamsollahi & Fatemizadeh (2022)* | MIT-BIH arrhythmia database | The Hidden Markov Model (HMM) | Time Series Features | Layered Hidden Markov Model (LHMM) | 97.10 |
| *Hosseinzadeh et al. (2021)* | MIT-BIH, EDB, AHA, CU, NSD, University of Toronto Dataset (UofTDB) | Multi-Class SVM | Time Domain Features, Frequency Domain Features Authentication Features | SVM + ANN | 95 |
| *Ahmad et al. (2021)* | MIT-BIH Arrhythmia database PTB Diabostic ECdataset | Multimodal Feature Fusion (MFF) | Frequency Domain Features | Multimodal Image (MIF) and Feature Fusion (MFF). | MIF: 98.4 MFF; 99.2 |
| *Essa & Xie (2021)* | MIT-BIH arrhythmia database | An Ensemble of DL-Based Model | RR frequencies, higher-order statistics (HOS) | CNN-LSTM and RRHOS-LSTM networks | CNN RRHOS:99.25 LSTM:95.81 |
| *Hou et al. (2019)* | MIT-BIH Arrhythmia Database | LSTM-Based Auto-Encoder | Beats-based Features Record based Features | SVM | 99.45 |
| *Li et al. (2022)* | MIT-BIH arrhythmia database | Bidirectional LSTM (BiLSTM) network | End to end | Bidirectional LSTM (BiLSTM) optimized Bayesian | 99.00% |
| *Ali, Kareem & Mohammed (2022)* | MIT-BIH and PTB Diagnostic db | 1D and 2D CNN model | Greatest Features | LSTM | 99.21 |

**Table 1** (*continued*)

| Researcher and year | Data sources | Method | Extracted features | Approaches to classification | Accuracy |
|---|---|---|---|---|---|
| *Jun et al. (2018)* | MIT-BIH and PTB Diagnostic db | 2D CNN model | Time Domain Features | 2-Dimentional CNN AlexNet VGGNet | 2D CNN:99.05 AlexNet: 98.5 VGGNet: 98.4 |
| *Cai et al. (2022)* | MIT-BIH and PTB Diagnostic db | 2D CNN model | Mix Time Series Features | ResNet50 + Dense Net | 93.7 |

**Table 2  Beats and classes of the heart's rate.**

| Heartbeat classes at AAMI | The rhythm subjects at MIT-BIH | Count of available beats |
|---|---|---|
| Normal beats (N) | N (Beats of normal) | 74,776 |
| | L (Left bundle branch of block) | 8,052 |
| | R (Right bundle branch of block) | 7,239 |
| | j (beats at escape nodal (junction)) | 229 |
| | e (beat of escape atrial escape ) | 16 |
| Beats of supraventricular (S) | A (atrial premature beat) | 2,528 |
| | S (premature supraventricular beat) | 2 |
| | J (premature nodal (junctional) beat) | 83 |
| Beats ventricular (V) | ! (ventricular flutter beat) | 472 |
| | V (ventricular premature contraction) | 7,115 |
| | E (beat of ventricular escape ) | 106 |
| | [ (ventricular flutter fibrillation start) | 6 |
| | ] (ventricular fibrillation flutter end) | 149 |
| Beats (F) | F (beats of ventricular fusion normal ) | 106 |
| Beats unknown (Q) | f (Fusion beats of paced and normal ) | 979 |
| | / (beat Paced ) | 7,001 |
| | Q (beats Unclassifiable ) | 33 |

to 2D beat images. CNN's excellent ability to recognize features automatically has made it one of the most widely utilized AI techniques (*Hassan et al., 2022*).

This study made use of data from the MIT-BIH arrhythmia database, which is a public standard dataset. It contains 48 separate 24-hour, two-channel ECG recordings, each lasting 30 min. Annotation of each file Heartbeat types are cataloged in the ATR file. According to ANSI/AAMI EC57:1998/(R) 2008, 18 unique original beat types are categorized as normal ectopic (N), unknown ectopic (Q), ventricular ectopic (V), and supraventricular ectopic (S), coded by 0, 1, 2, 3, and 4, respectively. The sample sizes utilized in this study are shown in Table 2 (*Rexy, Velmani & Rajakumar, 2021*).

There are 18 different kinds of beats, 14 of which have been categorized and 4 of which have not. Annotations of beats occur for all types of QRS waves on an electrocardiogram. As a result, many researchers and developers of QRS-detection tools utilize this database for research and development. Databases are often used to evaluate the performance of new software before it is implemented on devices used for various purposes. The analysis must be accurate; otherwise, the device selection will be wrong. In biomedical applications, such

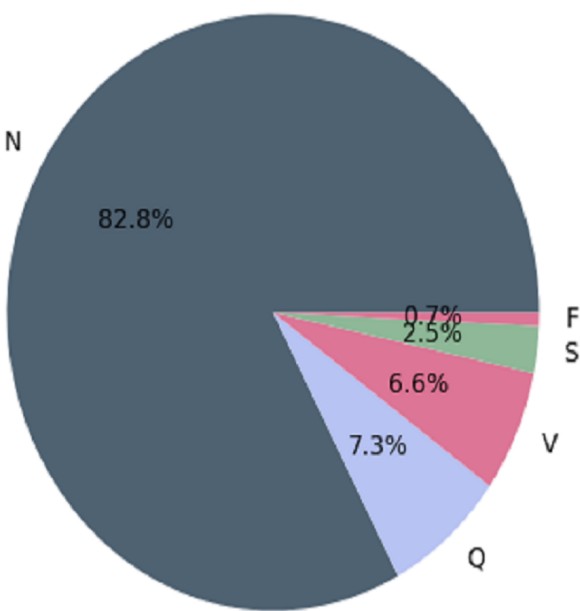

**Figure 1** **Architecture of the imbalanced databaset (*Guanglong, Xiangqing & Junsheng, 2019*).**

as QRS detection, which is crucial for many ECG monitoring devices, inaccurate results might make it more difficult for clinicians to establish a diagnosis and provide treatment. So, the doctor may make a more informed decision by searching the database for these applications (*Khalaf & Mohammed, 2021*).

## Balance of database with SMOTE

The MIT-BIH arrhythmia database contains several inconsistencies, with more "normal" classes than "abnormal" classes. When making predictions, classification algorithms may favor the majority class (more observations) over the minority class (fewer observations). In medical applications, misclassifying "abnormal" classes as "normal" can be fatal (*Guanglong, Xiangqing & Junsheng, 2019*). MIT-BIH has the following observations before balancing the arrhythmia database. This is depicted in Fig. 1.

### SMOTE

We used the Synthetic Minority Over-sampling Technique (SMOTE) as a resampling strategy to achieve balance in the MIT-BIH dataset. Oversampling the minority classes (SVEB (S), VEB (V), F, and Q) by generating synthetic instances minimizes the dataset's imbalance produced by the majority class (N). For each sample in the minority class, represented by a set of $k = 5$ synthetic instances, SMOTE assesses the differences between the sample and its five nearest neighbors (*Sarker, 2021*). These differences are then multiplied by a random number ranging from 0 to 1. The resampled MIT-BIH database reflects the AAMI-recommended arrhythmia categories: N, V, S, F, and Q. The dataset initially had imbalanced class distribution, with specific class counts of 72,471, 2,223, 5,788, 641, and 6,431. Through the application of SMOTE, we obtained a more balanced dataset with

**Table 3  Details of the MIT-BIH dataset before and after SMOTE.**

| Dataset | Classes | Classes before SMOTE | Classes after SMOTE |
| --- | --- | --- | --- |
| MIT-BIH DATASET | N (0) | 72,471 | 57,961 |
| | S (1) | 2,223 | 57,961 |
| | V (2) | 5,788 | 57,961 |
| | F (3) | 641 | 57,961 |
| | Q (4) | 6,431 | 57,961 |

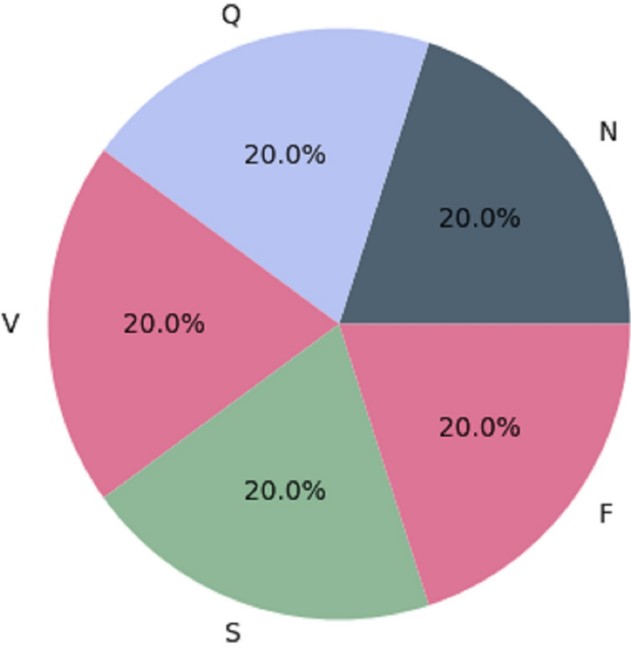

**Figure 2  Architecture of the balanced database (*Sarker, 2021*).**

57,961 instances, as detailed in Table 3. Figure 2 visually depicts the even distribution achieved through this resampling technique.

The balanced MIT-BIH dataset is now ready for additional preprocessing procedures to improve its usability for ML and DL algorithms. The following operations may involve data cleaning, addressing missing values, and splitting the dataset. These preprocessing steps are designed to refine the dataset, making it more structured, informative, and conducive to the best model performance.

### Categories of ECG signals

The MIT-BIH dataset provides a wide range of ECG signals that have been classified into separate classes, each indicating a particular cardiac state. The AAMI (Association for the Advancement of Medical Instrumentation) recommends the following categories: normal (N), supraventricular (S), ventricular ectopic (V), fusion beat (F), and unknown beat (Q).
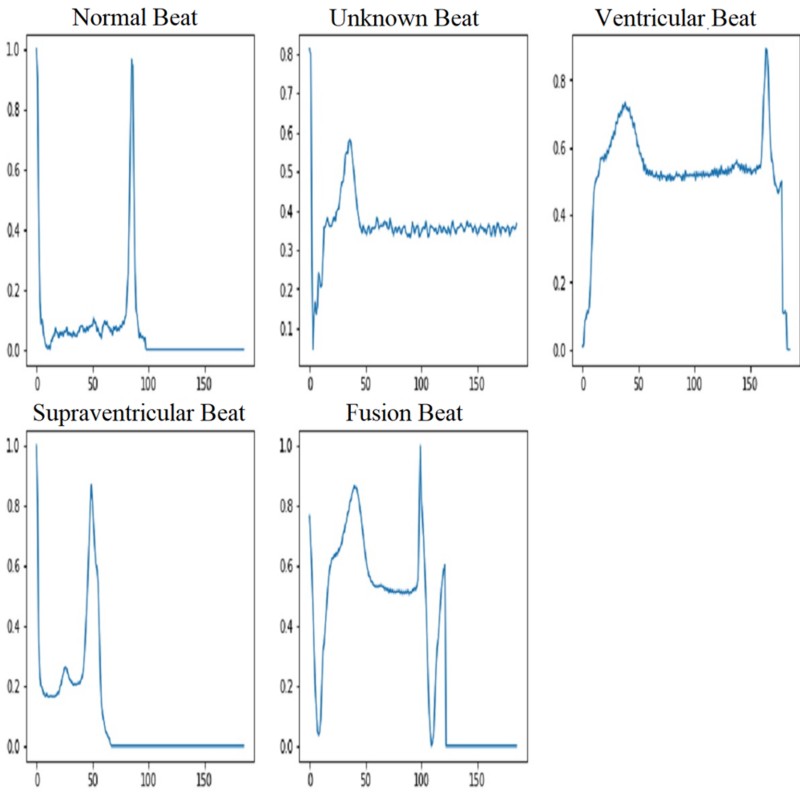

**Figure 3** **ECG signal samples of five classes (*Song et al., 2020*).**

The categorization of ECG signals in the MIT-BIH dataset facilitates targeted analysis and classification of various arrhythmias. The five types of ECG signals are depicted in Fig. 3.

## ECG data pre-processing

Preprocessing is performed on all ECG data to remove artifacts such as baseline drift, motion artifacts, muscle noise, and power line interference. Unstructured data is turned into a more understandable format during the data preparation phase. Before using ML and DL technologies, it is essential to ensure data integrity. To do this, we used an IIR filter to remove noise from the raw ECG signals.

## Importance of IIR Filter

Arrhythmias are categorized according to the time between R-peaks (RR intervals), abnormal P waves, and other physical characteristics. The QRS complex, an important part of the ECG signal that shows ventricular depolarization, was used in traditional arrhythmia classification studies to find problems with the heartbeat. *Manjula, Singh & Babu (2023)* proposed an optimized IIR filter for ECG signals categorized through QRS peak detection. They used the Pan-Tompkins algorithm to detect QRS complexes and calculate the RR intervals required for heart rate variability (HRV).

In addition to routine ECG readings, signals captured during physical activities may introduce artifacts and noise caused by body movements, which may reduce the accuracy

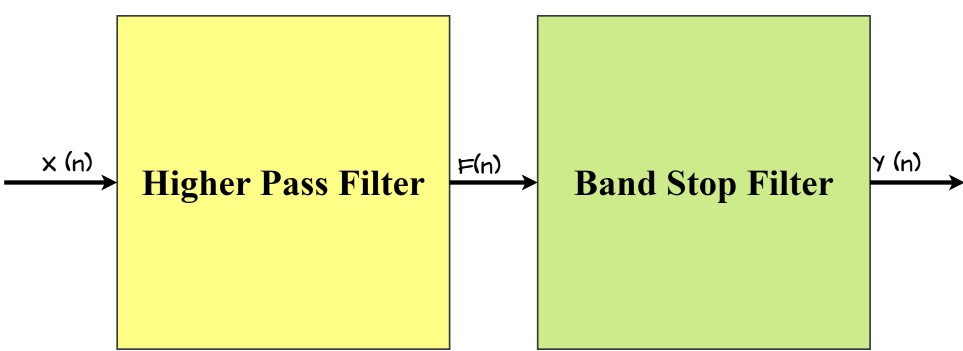

**Figure 4**  Architecture of proposed IIR Filter.

of QRS detection (*Apandi et al., 2022*). In that case, it is also possible to detect rhythm disorders by first filtering the ECG data with an IIR filter and then locating the QRS peak (*Amhia & Wadhwani, 2021*).

### Proposed IIR filter

The proposed IIR filter structure is a two-stage design that includes both a higher-pass filter and a stop-band filter, as illustrated in Fig. 4. This dual-stage architecture is intentionally adopted to achieve certain filtering objectives, ensuring effective noise reduction and feature extraction in the ECG signals.

Let $X(n)$ be the unfiltered ECG data input, and the IIR filter transfer function design definition is mentioned in Eq. (2).

$$Y(n) = X(n) * t1 * t2 \tag{1}$$

where t1 and ,t2 are transfer functions. In Eq. (3), the transfer function is shown.

$$X(z) = \sum_{n=-\infty}^{\infty} i^3 * X(n)Z^{-n}. \tag{2}$$

### Noise removal and feature extraction using the proposed IIR filter

An infinite impulse response (IIR) filter design consists of a high-pass filter (HPF) and a bandstop filter (BSF) that reduces noise and extracts specific features from electrocardiogram (ECG) signals. The HPF effectively eliminates baseline drift and low-frequency noise, both of which generate significant variations in ECG data. Its adjustable cutoff frequency increases filtering flexibility. Simultaneously, the BSF, which is defined by the quality factor (Q) and the center frequency, specifically targets specific frequencies, significantly reducing powerline interference. The use of a Butterworth filter for both the HPF and the BSF enables a consistent frequency response. The introduction of normalized cutoff frequencies improves the filters' adaptability across multiple sampling rates. Specifically, setting the HPF cutoff frequency to 0.5 Hz proves instrumental in effectively reducing noise in ECG data, further improving the overall

**Table 4  IIR filter parameters for extracting features from ECG signals.**

| Filter type | Cutoff frequency (Hz) | Order |
| --- | --- | --- |
| High-pass filter | 0.5 | 4 |
| Bandstop filter | 49.0–51.0 | 4 |

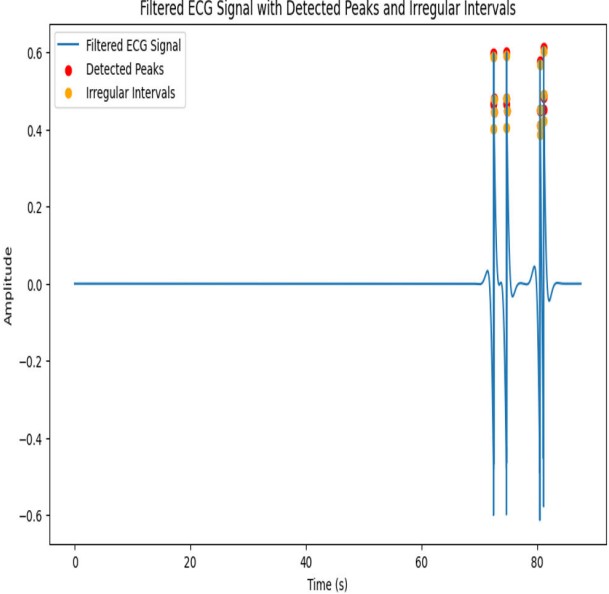

**Figure 5  Architecture of proposed IIR Filter.**

signal quality. Additionally, adding certain parameters to the Butterworth filter in the suggested IIR configuration improves feature extraction. Table 4 shows the values of these parameters.

After applying the given parameters, the filtered ECG signal with detected peaks and irregular intervals is shown in Fig. 5. A 103.39 beats per minute (BPM) pulse rate emphasizes the irregular intervals.

Thus, to improve the arrhythmia classification and make it easy to understand for both experts and non-experts, the ECG data is filtered using the proposed IIR filter. The filtered signals are input into the ML and DL models.

### Algorithm for IIR filter design
The proposed IIR filter algorithm is represented as follows:

---

**Algorithm 1** IIR Filter with High-pass and Band-stop Filters

---

**Require:** Raw ECG signal (*ecg_data*), Sampling frequency (*fs*), High-pass cut-off
frequency for Baseline Drift and Muscle Artifacts (*highpass_cutoff*), Filter order
for high-pass filter (*highpass_order*), Band-stop frequency range for Power Line
Interference and Artifact from Electronic Devices (*bandstop_range*), Filter order for
band-stop filter (*bandstop_order*)

**Ensure:** Filtered ECG signal (*filtered_ecg_data*)

1: Calculate normalized high-pass cut-off frequency: *normalized_cutoff* $\leftarrow$
   *highpass_cutoff* $/(0.5 \times fs)$

2: Design high-pass IIR filter using Butterworth filter design:

3: Determine filter coefficients using the `butter` function with 'highpass' type:
   *filter_coefficients* $\leftarrow$ butter(*highpass_order*, *normalized_cutoff*, 'highpass')

4: Apply high-pass IIR filter to the raw ECG signal:

5: Initialize *filtered_ecg_data* as an empty array.

6: Initialize filter state.

7: **for** each data point in *ecg_data*: **do**

8:     Apply high-pass IIR filter using *filter_coefficients*.

9:     Update filter state.

10:     Append the filtered data point to *filtered_ecg_data*.

11: **end for**

12: Calculate normalized band-stop frequencies:

13: *normalized_bandstop_low* $\leftarrow$ *bandstop_range*[0]$/(0.5 \times fs)$

14: *normalized_bandstop_high* $\leftarrow$ *bandstop_range*[1]$/(0.5 \times fs)$

15: Design band-stop IIR filter using Butterworth filter design:

16: Determine filter coefficients using the `butter` func-
    tion with 'bandstop' type: *bandstop_coefficients* $\leftarrow$
    butter(*bandstop_order*, [*normalized_bandstop_low*, *normalized_bandstop_high*], 'bandstop')

17: Apply band-stop IIR filter to the high-pass filtered ECG signal:

18: Initialize *filtered_ecg_data* as an empty array.

19: Initialize band-stop filter state.

20: **for** each data point in *filtered_ecg_data*: **do**

21:     Apply band-stop IIR filter using *bandstop_coefficients*.

22:     Update band-stop filter state.

23:     Append the filtered data point to *filtered_ecg_data*.

24: **end for**

25: **Output** *filtered_ecg_data* as the result of the combined high-pass and band-stop IIR
    filters applied to the raw ECG signal.

---

The term "baseline drift" refers to the low-frequency fluctuations observed in the electrocardiogram (ECG) signal that are unrelated to the heart's electrical activity. The potential cause of the issue may be identified as the positioning of the electrodes or other related conditions. It is greatly slowed down or stopped when an IIR high-pass filter is applied to a signal. It provides better signal quality and improves the clarity of the ECG signal. This makes identifying and analyzing the actual electrical activity of the heart easier, as it is free of interference from unrelated changes. Powerline interference is when random electrical signals get into ECG records through the power supply lines. This is called "interference from power lines". These unwanted signals, which usually come from electrical devices that use the same power grid, make noise in the ECG data, which could make the recorded heart signals less accurate and clear. To solve this problem, a stopband filter can be used to remove only the frequencies that are caused by powerline interference. This clears the ECG signal of this unwanted noise. Figure 6 shows five groups of filtered ECG signals, such as normal, supraventricular, ventricular, fusion, and unknown signals. All are filtered with a 150-Hz proposed IIR filter. It is very important for people who are not experts to understand what happens during the filtering process when they suggest an IIR filter for processing ECG signals, especially when it comes to classifying beats.

### Normal beat

By using an Infinite Impulse Response (IIR) high-pass filter first and then a low-pass filter, it is thought that the normal beat signal will effectively capture the ECG waveform that is usually associated with a healthy and normal heart rhythm. In a typical electrocardiogram (ECG) pattern, people can observe different waveforms known as P waves, QRS complexes, and T waves.

### Supraventricular beat

Electrical activity in the heart that occurs above the ventricles is what causes supraventricular beats, also known as supraventricular premature beats. This category includes a variety of abnormal heart contractions, including atrial premature contractions (APCs) and atrial fibrillation. By examining filtered supraventricular beats, it becomes clear that there are changes or problems with the electrocardiogram (ECG) waveform.

### Ventricular beat

The ventricles are the lower chambers of the heart that supply blood to the lungs and the rest of the body. Premature ventricular contractions (PVCs) are one of the most common types. In comparison to typical beats, the QRS complex in ventricular beats appears broad and irregular. Sometimes there is no P wave preceding myocardial depolarization. Ventricular beats can interrupt the regular rhythm of the heart, reducing cardiac output. The use of a filtered ventricular beat signal makes it easier to identify abnormalities in the QRS complex.

### Fusion beat

Fusion beats arise when both normal and abnormal electrical paths are active at the same time in the atria and ventricles. The filtered fusion beat signal may have a unique rhythm after the IIR high-pass and low-pass filters are applied to the ECG signal. This pattern

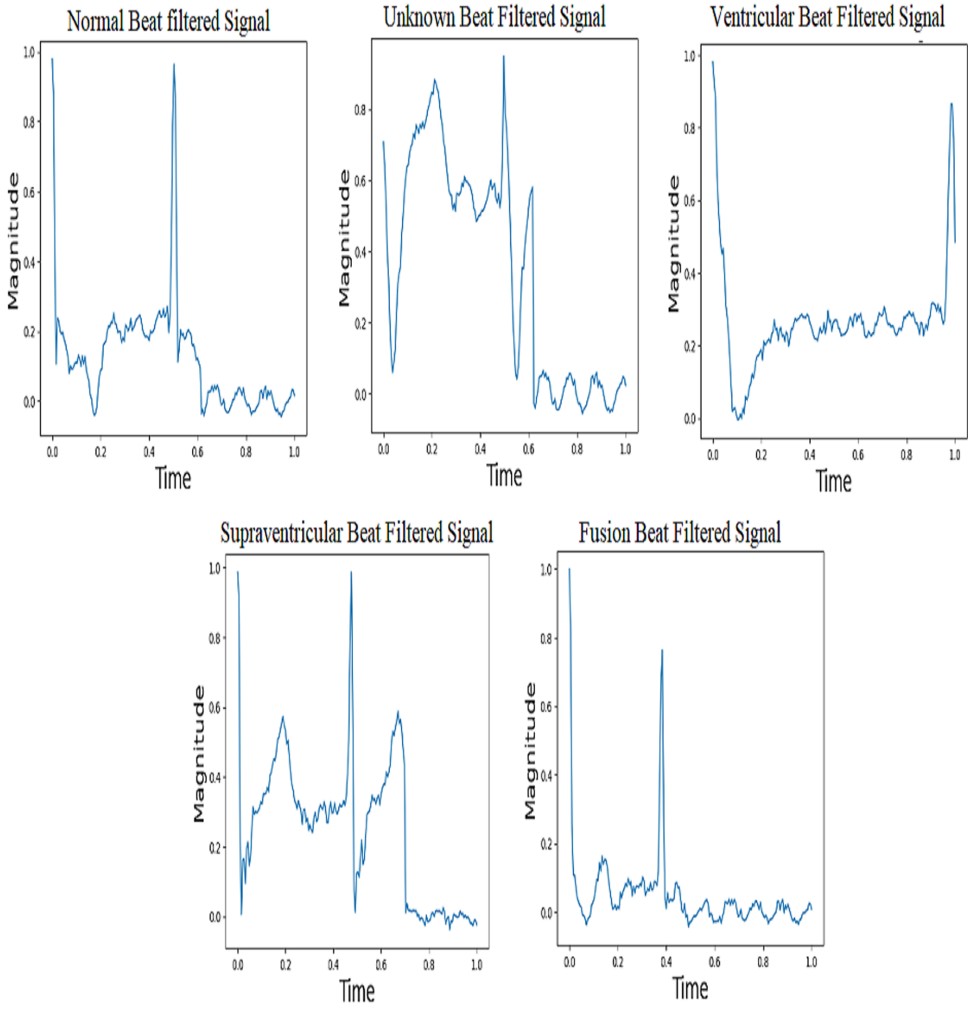

**Figure 6  Filtered ECG Signals with IIR filter.**

clearly indicates a mix of activities that occur between normal beats and abnormal beats (supraventricular or ventricular).

### Unknown beat

"Unknown beats" are typically ECG beats that cannot be accurately characterized as "normal", "superior ventricular", "ventricular", or "fusion". It is likely that these beats contain characteristics that make it difficult for computers or even skilled people to classify them. After applying the IIR filter to the filtered signal for unknown beats, changes that do not fit the standard patterns for recognized beat types may appear. It's possible that the pattern in the filtered unknown beat signal changes in some way from other beat kinds.

The IIR filtering approach makes it easier for non-experts to understand confusing ECG readings by removing redundant components. In practice, the elimination of baseline wandering is essential for accurate ECG interpretation. This promises that the observed differences are due to real heart activity and not to external factors.

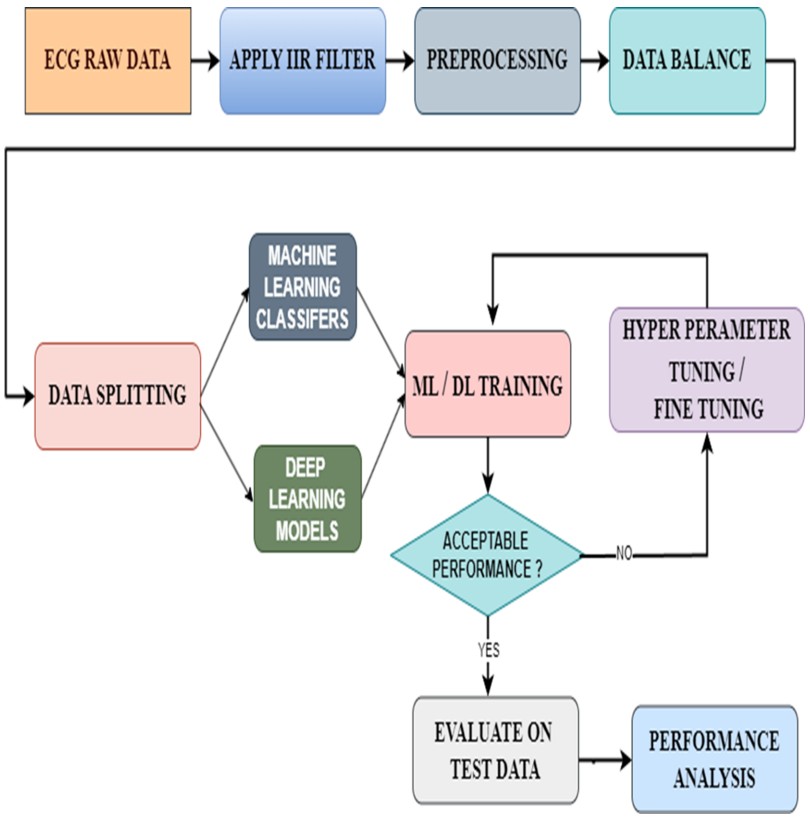

**Figure 7  Architecture of the proposed model.**

# PROPOSED METHODOLOGY

In this proposed model, we have first taken the data containing raw ECG signals from the MIT-BIH database. Then we filtered the signals with an IIR filter. The best of the filtered signals were preprocessed. In this preprocessing, we check whether there are any missing data or NAN values and add them to the data balance. Data balancing is a very important process because feeding the correct data to the ML or DL models will result in accurate results. For this, we used the Smote Technique. Thus, from the data balance, the balanced data is given to the data split to split it into training and testing. Later, we gave the split bata to the ML and DL models for arrhythmia classification. However, this study not only used pre-existing models but also enhanced their functionalities using new methodologies. The ML classifiers used the hyperparameter tuning technique to achieve this, whereas the DL models used the fine-tuning technique. Then a performance analysis was done to examine the results from all ML and DL models. Densenet-121 has given good results in this area. We have included the proposed model that includes this entire process in Fig. 7.

## Hyperparameter tuning for ML classifiers

Hyperparameter-tuning significantly improved the performance of the ML models used to classify ECG signals in this study. A full grid search was performed, looking at a range of

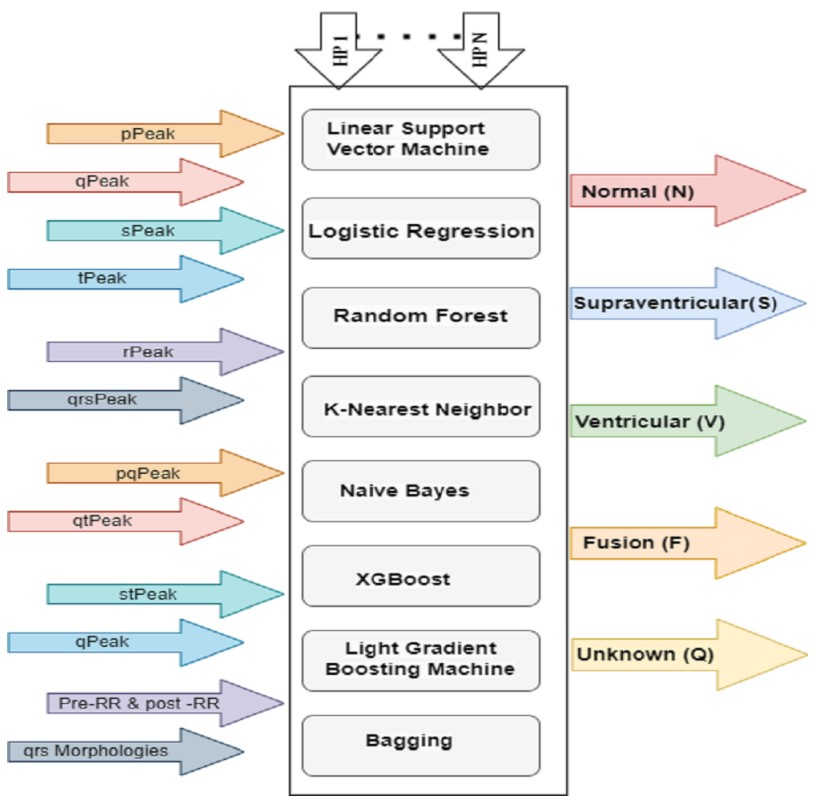

**Figure 8** HP architecture for ML classifiers.

hyperparameter values denoted by the letters h1 through hn and examining five different types of ECG signals. There are also pre- and post-RR forms, with peaks and intervals at P, Q, R, S, and T. Eight machine learning models are depicted in Fig. 8 depicts eight ML classifiers, including decision line support vector machines (LSVM), logistic regression (LR), random forest (RF), K-nearest neighbor (KNN), naive bayes (NB), XGBoost (XGB), light gradient boost machines (LGBM), and bagging (BAGG). All of these models need tweaks to their hyperparameters.

## Fine tuning for DL models

Fine-tuning significantly improved the performance of the DL models used to categorize Arrhythmias. The pre-trained models AlexNet, VGG-16, ResNet-50, and DenseNet-121 are fine-tuned with parameters such as ADAM optimizer, maximum pooling, minimum batch size of 128, the learning rate of 0.01, and drop factor of 0.05 for the best accuracy. Fig. 9 depicts the various model training elements that are used to fine-tune a model so that it performs optimally. It was observed how the speed changes when the batch size changes. It takes less time to train with a large batch size, but it is easy to overfit. It takes longer to train with a small batch size, but the results are better.

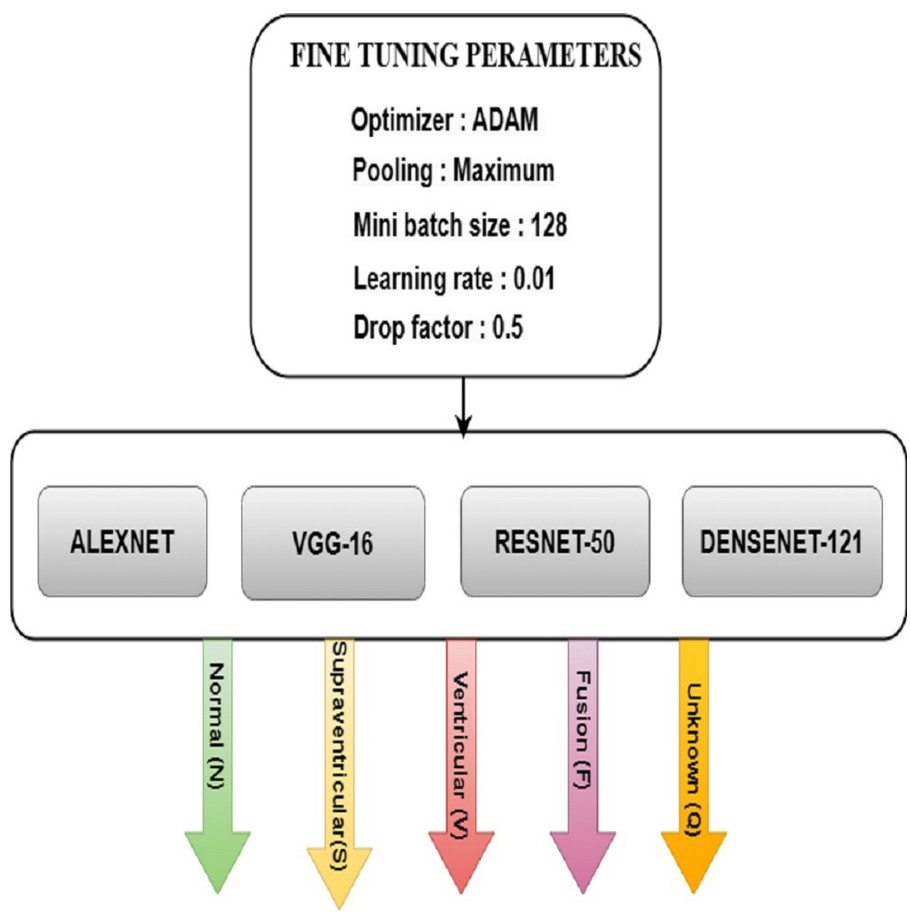

**Figure 9** **FT architecture for ML models.**

# IMPLIMENTED DIFFERENT MODELS RESULTS AND ANALYSIS

This section starts with metrics for performance evaluation. This section follows the machine learning results with the deep learning outcomes. Finally, the suggested technique's efficiency findings are shown, along with comparisons to other network results.

## Performance evaluation metrics

Evaluating an arrhythmia classification model involves key metrics like sensitivity, specificity, accuracy, precision, and F1 score. Sensitivity focuses on minimizing false negatives (FN), specificity reduces false positives (FP), accuracy considers overall correctness, precision minimizes false positives, and the F1 score provides a balanced view. Regular monitoring of these metrics is crucial for the model's reliability in clinical applications. The formulas for calculating sensitivity, specificity, accuracy, precision, and F1 score are in Eqs. (4), (5), (6) and (7).

$$\text{Sensitivity}(S_E) = \frac{T_p * 100}{T_p + F_n} \tag{3}$$

$$\text{Specificity } (S_P) = \frac{T_N * 100}{T_N + F_P} \tag{4}$$

$$\text{Acuuracy} = \frac{(T_N + T_P) * 100}{T_N + F_P + T_P + F_N} \tag{5}$$

$$\text{Precision} = \frac{TP * 100}{TP + FP} \tag{6}$$

$$F1\text{Score} = 2 * \frac{(P * S_E)}{P + S_E}. \tag{7}$$

## The final results of both ML and DL

After filtering regular ECG signals through the proposed IIR filter, this work utilizes ML and DL models to perform distinct evaluations on the filtered signals. Consequently, the findings derived from the ML and DL methodologies have been included in this sub-section.

The IIR filter has the purpose of passing or reducing certain frequency components of the ECG signal. This is important in ECG analysis because it helps users concentrate on particular areas of the signal associated with heart activity while filtering out unnecessary noise or interference. The IIR filter can efficiently lower high-frequency noise generated through muscle action or external interference. This contributes to a more accurate and trustworthy description of the underlying heart activity. The IIR filter may change the amplitude and phase of the ECG signal. This may help to improve particular qualities of interest or align the signal for better analysis. The IIR filter can produce smoothing or sharpening effects on the signal, depending on its design. Smoothing reduces the effect of little variations while sharpening focuses on sudden fluctuations in the signal. IIR filters may cause phase distortions, which might be essential in situations where signal timing is vital. Understanding and regulating phase response is critical for retaining the ECG waveform's temporal features.

## Quantitative evaluation of eight machine learning classifications analysis

Precision evaluates the accuracy of positive predictions, which is an important metric. It is the proportion of actual positives to true positives plus false positives. In medical diagnostics, for instance, greater precision suggests a lower probability of misclassifying a healthy person as having a problem. Recall is a measure of an arrhythmia classification model's capacity to identify all actual classes of arrhythmias. It is the ratio of true positives to the sum of true positives and false negatives, which is essential to avoiding overlooking possible cases. High recall ensures complete identification of arrhythmias, reducing the number of missing significant instances. The F1 score finely balances accuracy and recall, allowing for a thorough model evaluation. This is especially beneficial when dealing with

**Table 5  LSVM without HP.**

| Class | Precision | Recall | $f$1-score |
|---|---|---|---|
| 0 | 0.95 | 0.88 | 0.92 |
| 1 | 0.92 | 0.97 | 0.95 |
| 2 | 0.97 | 0.96 | 0.95 |
| 3 | 0.98 | 1.00 | 0.99 |
| 4 | 0.97 | 0.98 | 0.98 |

**Table 6  LR without HP.**

| Class | Precision | Recall | $f$1-score |
|---|---|---|---|
| 0 | 0.89 | 0.90 | 0.92 |
| 1 | 0.92 | 0.97 | 0.95 |
| 2 | 0.91 | 0.90 | 0.91 |
| 3 | 0.91 | 0.99 | 0.97 |
| 4 | 0.98 | 0.99 | 0.99 |

**Table 7  RF without HP.**

| Class | Precision | Recall | $f$1-score |
|---|---|---|---|
| 0 | 0.95 | 0.88 | 0.92 |
| 1 | 0.93 | 0.97 | 0.95 |
| 2 | 0.96 | 0.95 | 0.96 |
| 3 | 0.96 | 1.00 | 0.98 |
| 4 | 0.99 | 0.99 | 0.99 |

uneven class distributions of false positives and false negatives. In arrhythmia cases, a higher F1 score indicates a well-balanced precision–recall trade-off, indicating model strength.

The performance analysis in terms of precision, recall, and F1 score of each and individual classifiers related to machine learning such as LSVM, LR, RF, KNN, NB, XGB, LGBM and BAGG s without HP is shown from Tables 5, 6, 7, 8, 9, 10, 11 and 12.

The performance analysis in terms of precision, recall, and F1 score of each and individual classifiers related to machine learning such as LSVM, LR, RF, KNN, NB, XGB, LGBM and BAGG s with HP is shown in Tables 13, 14, 15, 16, 17, 18, 19 and 20.

## Analysis of the quantitative evaluation of four deep learning models

The accuracy curve in the training shows how model accuracy changes over epochs or iterations. The loss curve illustrates the error or loss function of the model throughout training epochs. Confusion matrix tables are summarized for the performance of categorization models. It displays true positives and negatives, as well as false positives and negatives.

The deep learning-related model's accuracy and loss without the fine-tuning of four different methods such as AlexNet, VGG-16, ResNet-50, and DenseNet-121 is displayed from Figs. 10, 11, 12 and 13 and with fine-tuning from Figs. 14, 15, 16 and 17. As displayed

**Table 8 KNN without HP.**

| Class | Precision | Recall | $f$1-score |
|---|---|---|---|
| 0 | 0.58 | 0.64 | 0.61 |
| 1 | 0.83 | 0.69 | 0.75 |
| 2 | 0.75 | 0.79 | 0.77 |
| 3 | 0.85 | 0.89 | 0.87 |
| 4 | 0.94 | 0.90 | 0.92 |

**Table 9 NB without HP.**

| Class | Precision | Recall | $f$1-score |
|---|---|---|---|
| 0 | 0.45 | 0.12 | 0.18 |
| 1 | 0.90 | 0.13 | 0.23 |
| 2 | 0.58 | 0.21 | 0.31 |
| 3 | 0.81 | 0.76 | 0.78 |
| 4 | 0.29 | 0.99 | 0.45 |

**Table 10 XGB without HP.**

| Class | Precision | Recall | $f$1-score |
|---|---|---|---|
| 0 | 0.98 | 0.95 | 0.97 |
| 1 | 0.98 | 0.99 | 0.99 |
| 2 | 0.98 | 0.99 | 0.99 |
| 3 | 0.99 | 1.00 | 0.99 |
| 4 | 1.00 | 0.99 | 1.00 |

in Figs. 10, 11, 12 and 13, among the four different models the DenseNet-121 has achieved the highest accuracy at 99% when compared to the performance of the remaining three models. As displayed in Figs. 14, 15, 16 and 17, among the four different models once again DenseNet-121 achieved the highest accuracy at 99.97% compared to the performance of the remaining three models and proved to be a better model.

## Comparison of ML classification techniques

We analyze the performance of numerous machine learning classifiers available for this study, as well as their performance with hyperparameter tuning when applied to filtered ECG data using the FIR filter. The outcomes are summarised in the Table 21 below. Fig. 18 illustrates the variations in HP (hyperparameter tuning) values across all classifiers.

## Comparison of DL models

As mentioned above, the accuracy results obtained from deep learning models through the FT (fine tuning) technique are included in the following Table 22. Figure 19 shows how the FT values differentiate between the various DL models.

**Table 11  LGBM without HP.**

| Class | Precision | Recall | ƒ1-score |
|---|---|---|---|
| 0 | 0.97 | 0.94 | 0.95 |
| 1 | 0.97 | 0.99 | 0.98 |
| 2 | 0.98 | 0.98 | 0.98 |
| 3 | 0.99 | 1.00 | 0.99 |
| 4 | 1.00 | 0.99 | 1.00 |

**Table 12  BAGG without HP.**

| Class | Precision | Recall | ƒ1-score |
|---|---|---|---|
| 0 | 0.97 | 0.95 | 0.96 |
| 1 | 0.98 | 1.00 | 0.99 |
| 2 | 0.98 | 0.98 | 0.98 |
| 3 | 0.99 | 1.00 | 0.99 |
| 4 | 1.00 | 0.99 | 0.99 |

**Table 13  LSVM with HP.**

| Class | Precision | Recall | ƒ1-score |
|---|---|---|---|
| 0 | 0.96 | 0.89 | 0.92 |
| 1 | 0.95 | 0.99 | 0.97 |
| 2 | 0.95 | 0.97 | 0.96 |
| 3 | 0.98 | 1.00 | 0.99 |
| 4 | 0.98 | 0.99 | 0.99 |

## THE PERFORMANCE DISCUSSION OF THE IIR FILTER IN COMPARISON TO FILTERS OF OTHER TYPES

In this study, we propose that applying a filter may provide completely distinct and more accurate results when classifying arrhythmias than would be possible without a filter. In this method, we used the IIR filter. The reason for the advantage of IIR filters over ordinary filters is that IIR filters generally require fewer coefficients to perform similar filtering operations, run faster, and require less memory space.

Let us examine those researchers who have used other filters, apart from the Infinite Impulse Response (IIR) filter, for the purpose of rhythmic categorization. A technique for the automated categorization of electrocardiograms (ECG) using a combination of several support vector machines (SVMs) was suggested by *Mondéjar-Guerra et al. (2019)*. The accuracy rate was 94.5% with high-frequency noise filtering. *Mathews, Kambhamettu & Barner (2018)* showed how the Restricted Boltzmann Machine (RBM) and deep belief networks (DBN) can be used in the real world to classify electrocardiograms (ECGs). They used a bandpass filter and achieved an accuracy of 75.5% in their classification work.

*Raj & Ray (2018)* introduced a novel approach for feature extraction by using the sparse representation methodology. This method effectively represents various electrocardiogram

**Table 14  LR with HP.**

| Class | Precision | Recall | ƒ1-score |
|---|---|---|---|
| 0 | 0.97 | 0.98 | 0.97 |
| 1 | 0.99 | 0.99 | 0.99 |
| 2 | 0.99 | 0.99 | 0.99 |
| 3 | 0.99 | 1.00 | 1.00 |
| 4 | 1.00 | 0.99 | 1.00 |

**Table 15  RF with HP.**

| Class | Precision | Recall | ƒ1-score |
|---|---|---|---|
| 0 | 0.95 | 0.88 | 0.92 |
| 1 | 0.93 | 0.97 | 0.95 |
| 2 | 0.96 | 0.95 | 0.96 |
| 3 | 0.96 | 1.00 | 0.98 |
| 4 | 0.99 | 0.98 | 0.99 |

**Table 16  KNN with HP.**

| Class | Precision | Recall | ƒ1-score |
|---|---|---|---|
| 0 | 0.62 | 0.70 | 0.66 |
| 1 | 0.89 | 0.71 | 0.79 |
| 2 | 0.76 | 0.82 | 0.79 |
| 3 | 0.85 | 0.89 | 0.87 |
| 4 | 0.95 | 0.91 | 0.93 |

(ECG) signals using a band-pass filter, achieving an accuracy of 90.3%. *Wang et al. (2020)* presented a dual, fully connected neural network model for accurate classification of heartbeats using a notch filter with an impressive accuracy rate of 93.4%. *Dias et al. (2021)* suggested that single-lead ECG data could be used to classify arrhythmias with an accuracy of 88.6% by using the inter-patient paradigm and a band-pass filter. Using a band-stop filter and a deep neural network, *Wu et al. (2022)* suggested a classifier with an accuracy of 91.9% for automatically identifying arrhythmias. The Table 23 below contains every relevant piece of information related to these challenges.

In this research article, we have implemented both the techniques of ML and DL together. Further with these two, we have performed the implementation on a total of 12 models. such as in ML and DL . Among these 12 proposed models, the best efficiency and performance is achieved by DenseNet-121 compared to the remaining models as displayed in Table 24. Based on these results, it appears that the DenseNEt-121 method is reliable for automatically categorizing cardiac arrhythmia. Using established CNN architectures instead of building a deep CNN from scratch provides a reliable detection method.

**Table 17  NB with HP.**

| Class | Precision | Recall | ƒ1-score |
|---|---|---|---|
| 0 | 0.60 | 0.51 | 0.55 |
| 1 | 0.66 | 0.60 | 0.63 |
| 2 | 0.70 | 0.61 | 0.65 |
| 3 | 0.68 | 0.91 | 0.78 |
| 4 | 0.87 | 0.89 | 0.88 |

**Table 18  XGB with HP.**

| Class | Precision | Recall | ƒ1-score |
|---|---|---|---|
| 0 | 0.99 | 0.96 | 0.98 |
| 1 | 0.98 | 0.99 | 0.99 |
| 2 | 0.99 | 0.99 | 0.99 |
| 3 | 0.99 | 1.00 | 1.00 |
| 4 | 0.99 | 1.00 | 1.00 |

**Table 19  LGBM with HP.**

| Class | Precision | Recall | ƒ1-score |
|---|---|---|---|
| 0 | 0.98 | 0.95 | 0.97 |
| 1 | 0.97 | 0.99 | 0.98 |
| 2 | 0.99 | 0.99 | 0.99 |
| 3 | 0.99 | 1.00 | 1.00 |
| 4 | 0.99 | 1.00 | 0.99 |

## CONCLUSION

In this detailed article, we examined arrhythmia classification for non-experts, with a focus on using the IIR filter in machine learning and deep learning models. We investigated data from eight machine learning classifiers and four deep learning models to correctly detect arrhythmias from ECGs. According to our findings, DenseNet-121 was evidently the best of the classifiers and models we tested, with an incredible 99% accuracy without FT and 99.97% accuracy with FT. This important study is notable since it simplifies arrhythmia classification for individuals who aren't experts in the field. The amazing performance of DenseNet-121 can be related to the time-consuming process of hyperparameter adjusting and fine-tuning. We discovered the model's perfect area by carefully examining and modifying its hyperparameters and minute details. This includes adjusting network variables such as learning rate, batch size, number of layers, and depth.

The outstanding performance can be attributed to a mix of rigorous hyperparameter tuning, early pausing, and data augmentation. These optimization efforts considerably improved the model's predictive capacity by increasing its accuracy and recall. These findings have far-reaching implications for the advancement of healthcare information technology as a whole. The consequences of proper arrhythmia classification, especially

**Table 20   BAGG with HP.**

| Class | Precision | Recall | ƒ1-score |
|---|---|---|---|
| 0 | 0.98 | 0.95 | 0.96 |
| 1 | 0.98 | 0.99 | 0.99 |
| 2 | 0.98 | 0.99 | 0.98 |
| 3 | 0.99 | 1.00 | 1.00 |
| 4 | 0.99 | 0.99 | 0.99 |

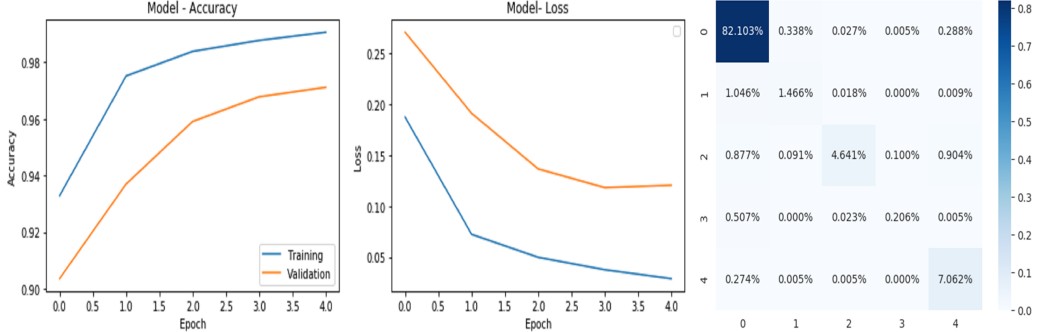

**Figure 10   AlexNet without FineTune.**

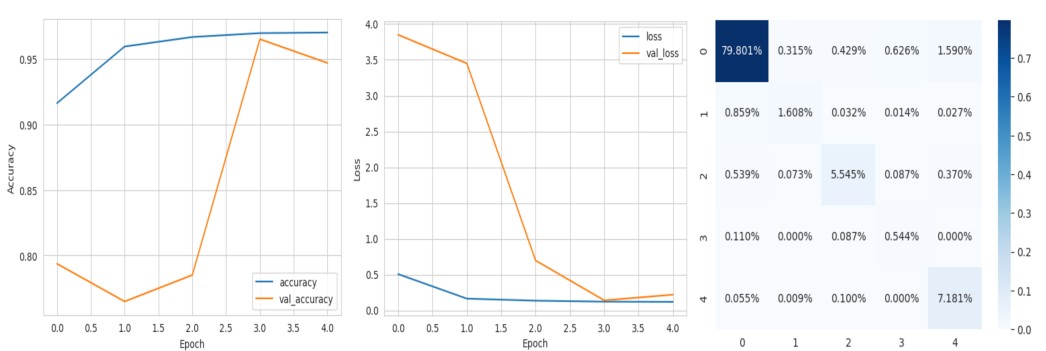

**Figure 11   VGG-16 without FineTune.**

when conducted by non-specialists, are significant. This not only improves access to healthcare in general, but it also ensures the early detection of cardiac illnesses that may otherwise be deadly. DenseNet-121 has shown promising outcomes, so there is optimism that it will be extensively employed in healthcare. This research also provides a potential route for further exploration. DenseNet-121 is at the vanguard of this study's celebration of deep learning models' capabilities. Additional studies might focus on polishing the model even further, making it more robust, and taking into consideration real-world healthcare applications. This last paragraph fully discusses the role of hyperparameter adjusting in increasing model performance.

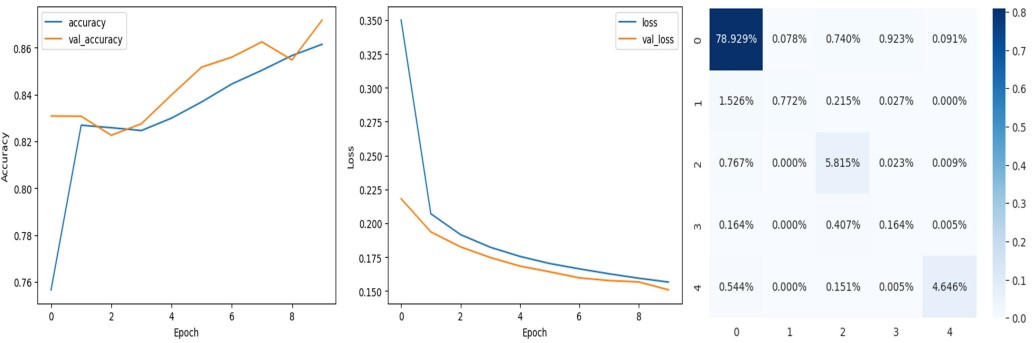

**Figure 12  ResNet-50 without FineTune.**

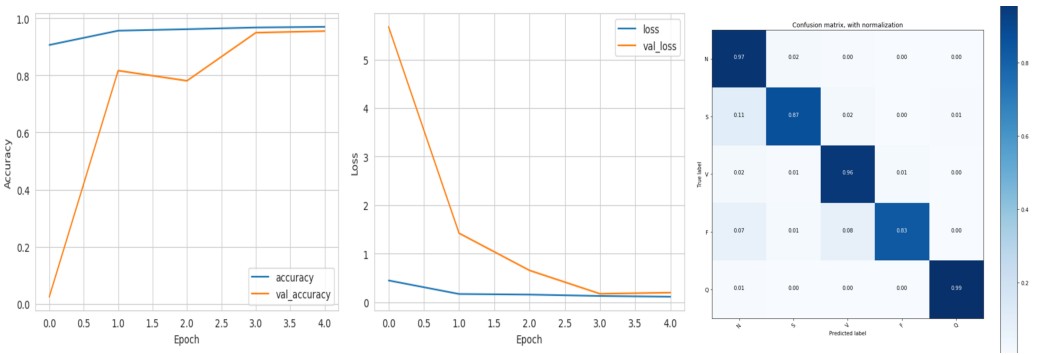

**Figure 13  DenseNet-121 without FineTune.**

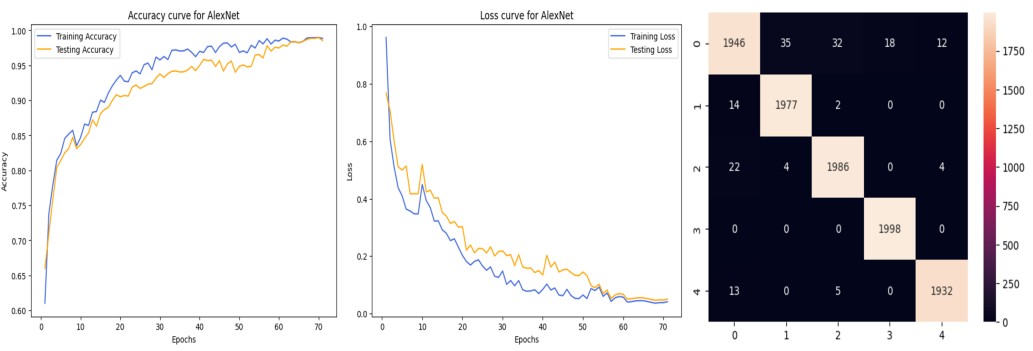

**Figure 14  AlexNet with FineTune.**

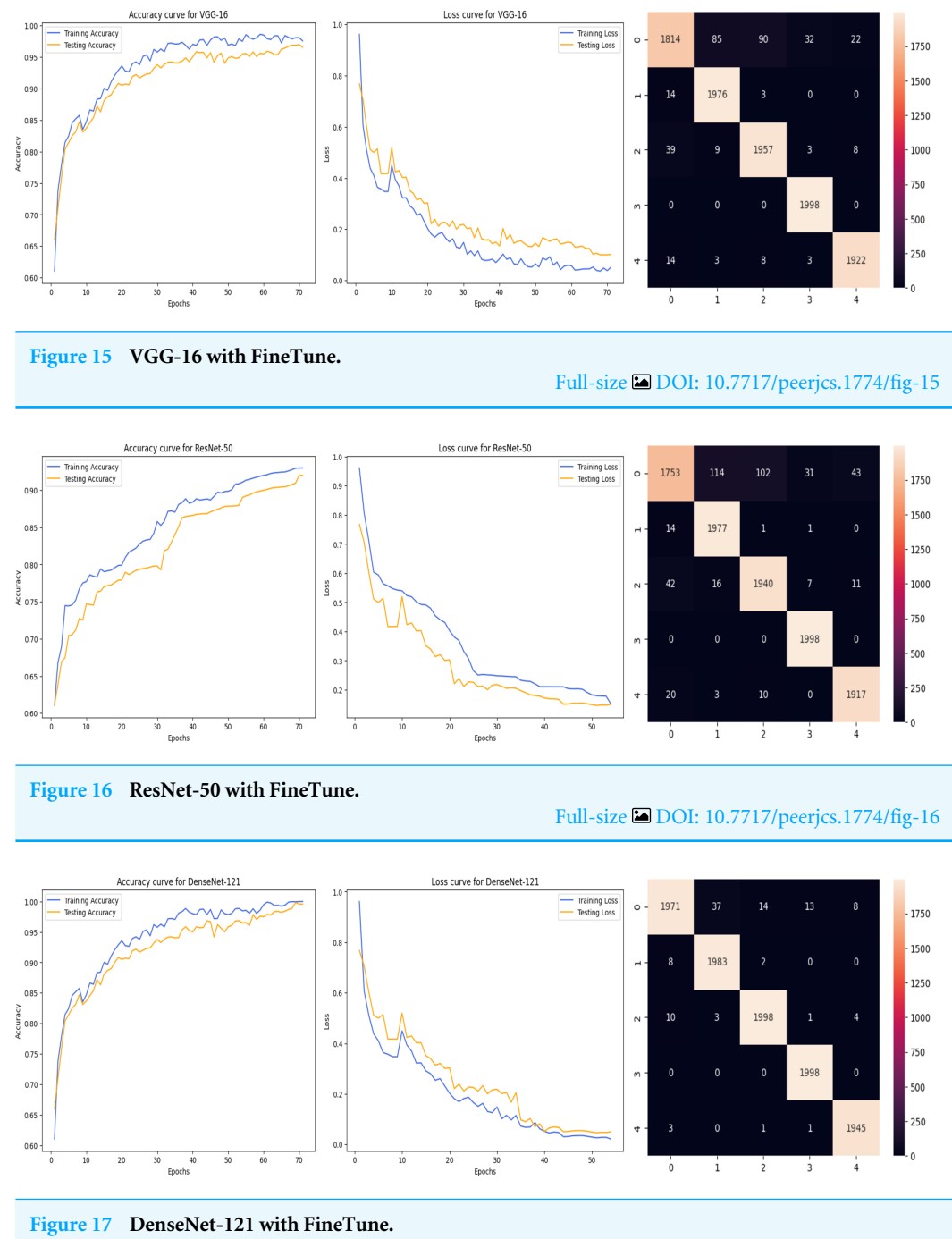

**Figure 15** VGG-16 with FineTune.

**Figure 16** ResNet-50 with FineTune.

**Figure 17** DenseNet-121 with FineTune.

## Feature scope

We plan to deepen the network and see if the model can pick up features that are applicable to any ECG dataset, to see how far we can push this effort.

**Table 21  ML classifiers accuracy comparison without HP and with HP.**

| S.no | ML classifier | Accuracy without HP | Accuracy with HP |
|------|---------------|---------------------|------------------|
| 1 | Linear SVM (LSVM) | 96% | 98% |
| 2 | Logistic Regression (LR) | 90% | 97% |
| 3 | Random Forest (RF) | 96% | 97% |
| 4 | K-Nearest Neighbor (KNN) | 78% | 85% |
| 5 | Naive Bayes (NB) | 44% | 70% |
| 6 | XGBoost (XGB) | 95% | 97% |
| 7 | Light GBM (LGBM) | 96% | 98% |
| 8 | Bagging (BAGG) | 96% | 98% |

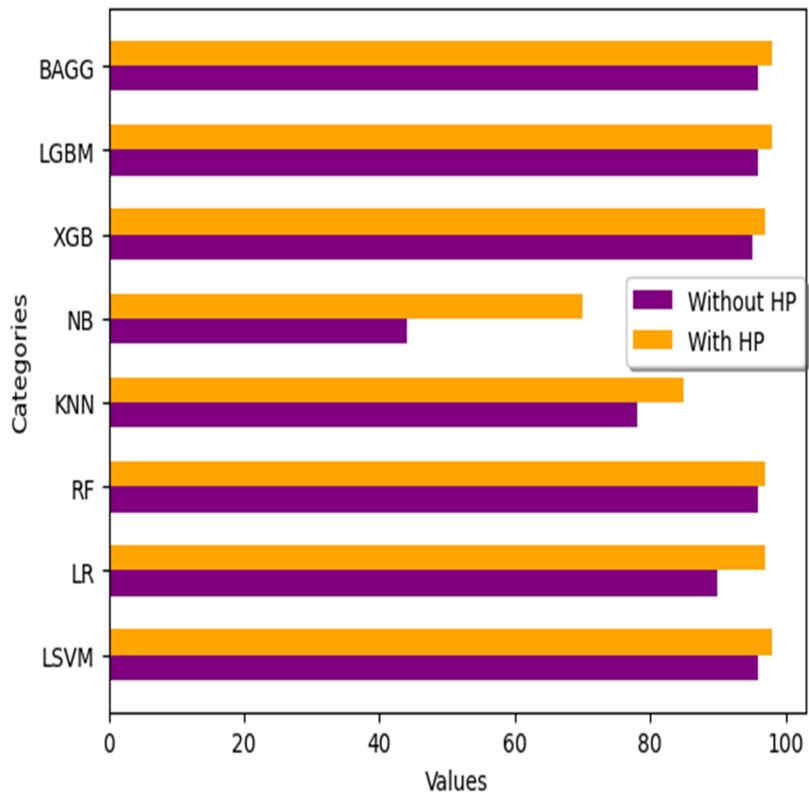

**Figure 18  Accuracy comparison with different ML classifiers.**

**Table 22   DL models accuracy comparison without and with FT.**

| S. no | DL classifier | Accuracy without FT | Accuracy with FT |
|---|---|---|---|
| 1 | AlexNet | 96.37% | 98.23% |
| 2 | VGG-16 | 94% | 97.04% |
| 3 | ResNet-50 | 87.2% | 92.1% |
| 4 | DenseNet-121 | 99% | 99.97% |

**Figure 19   Accuracy comparison with different DL models.**

**Table 23   Comparison of the performance with the existing filter methods.**

| Ref | Type of filter | Accuracy |
|---|---|---|
| *Mondéjar-Guerra et al. (2019)* | SVM with FIR Filter | 94.5% |
| *Mathews, Kambhamettu & Barner (2018)* | DBN with Band-Pass Filter | 75.5% |
| *Raj & Ray (2018)* | CNN with Band-Pass filter | 90.3% |
| *Wang et al. (2020)* | CNN with Notch filter | 93.4% |
| *Dias et al. (2021)* | DNN with Band-Pass Filter | 88.6% |
| *Wu et al. (2022)* | DNN with Band-Stop Filter | 91.9% |
| Model name | DenseNet-121 with IIR filter | 99.97% |

**Table 24 Accuracy comparison of 12 proposed models.**

| S. no | ML and DL Models | Accuracy |
|---|---|---|
| 1 | Linear SVM (LSVM) | 98% |
| 2 | Logistic Regression (LR) | 97% |
| 3 | Random Forest (RF) | 97% |
| 4 | K-Nearest Neighbor (KNN) | 85% |
| 5 | Naive Bayes (NB) | 70% |
| 6 | XGBoost (XGB) | 97% |
| 7 | Light GBM (LGBM) | 98% |
| 8 | Bagging (BAGG) | 98% |
| 9 | AlexNet | 98.23% |
| 10 | VGG-16 | 97.04% |
| 11 | ResNet-50 | 92.1% |
| 12 | DenseNet-121 (Without FT) | 99% |
| 13 | DenseNet-121(with FT) | 99.97% |

### Funding
The authors received no funding for this work.

### Competing Interests
The authors declare there are no competing interests.

### Author Contributions
- Mallikarjunamallu K conceived and designed the experiments, performed the experiments, analyzed the data, performed the computation work, prepared figures and/or tables, authored or reviewed drafts of the article, iir filter apply to the raw ecg signals, and approved the final draft.
- Khasim Syed conceived and designed the experiments, performed the experiments, analyzed the data, performed the computation work, prepared figures and/or tables, authored or reviewed drafts of the article, and approved the final draft.

### Data Availability
The code is available in the Supplemental File and at GitHub:

-https://github.com/kmmallu/Arrhythmia-Classification.

- kmmallu. (2023). kmmallu/Arrhythmia-Classification: Initial Release (0.1.0). Zenodo. https://doi.org/10.5281/zenodo.10361994.

The dataset uses features extracted two-lead ECG signal (lead II, V) from the MIT-BIH Arrhythmia dataset (Physionet) at Kaggle:

https://www.kaggle.com/datasets/shayanfazeli/heartbeat.

## Supplemental Information

Supplemental information for this article can be found online at http://dx.doi.org/10.7717/peerj-cs.1774#supplemental-information.

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
