# Peer review of "Arrhythmia classification for non-experts using infinite impulse response (IIR)-filter-based machine learning and deep learning models of the electrocardiogram"

_PeerJ Computer Science, doi:10.7717/peerj-cs.1774_

## Round 0.1 · original submission · Major Revisions

The study looks interesting but reviewers raised a number of issues that need to be fixed before this article can be considered for acceptance. Please address these points and prepare a new version.

**Language Note:** PeerJ staff have identified that the English language needs to be improved. When you prepare your next revision, please either (i) have a colleague who is proficient in English and familiar with the subject matter review your manuscript, or (ii) contact a professional editing service to review your manuscript. PeerJ can provide language editing services - you can contact us at copyediting@peerj.com for pricing (be sure to provide your manuscript number and title). – PeerJ Staff

·

Basic reporting

In this manuscript, they construct an arrhythmia classification model using well-known machine learning algorithms on a worn-out public electrocardiogram dataset. In comparison to numerous previous studies that conducted similar investigations, there is no novel insight included, rendering the paper scientifically meaningless. Do they want to discuss the usefulness of noise filters in this paper, or is it about the performance of machine learning?

Experimental design

It is evident at a glance that the content is of poor quality when they extensively explain the mechanisms of commonly known machine learning models in the methodology section.

Validity of the findings

The reviewer not only questions the validity of these research findings but also the validity of the research objectives.

Additional comments

There have been countless attempts to classify electrocardiogram waveforms using machine learning. Research involves forming hypotheses about new ideas and then validating them.

Reviewer 2 ·

Basic reporting

The logic of the paragraph needs improvement.

In line 17, if the quoted line is from a reference, please cite.
In line 28, please indicate on what indicator is 99% in.
line 31, please explain what is HD, and what is AHA. The line `. The AHA says American Indians have the highest rate of heart disease` doesn't seem to be related with the rest of the article.
Line 36, please cite the source of "millions of people..."

Line 66: We work based on -> Our work is based on. Remove the period after the word "system"
Line 69 what does four29 mean
Line 99, Deep learning is much better than machine learning, please cite from any resources this conclusion is from.
Line 105, remove the space before the coma

line 117 what is 0.1

line 136 ECg -> ECG

Experimental design

Table 2 is hard to read
Line 137- line 146, please use algorithm template rather than bullet points for algorithms
Figure 2 caption please add a description for each chart.
Line 197 please explain DT

In line 28, should also include your proposed model.
It's commendable that the author gives the insight of the distribution of the dataset.

Validity of the findings

The experiment was well constructed and compared some known models in this task.


The novelty should be improved.

Reviewer 3 ·

Basic reporting

The paper is generally well-written in clear, professional English. It provides a clear introduction and background, setting the context for the study. The introduction effectively highlights the importance of arrhythmia classification and the challenges associated with it. The paper references relevant literature, providing a solid foundation for the research. The structure adheres to standards and is generally clear, making it easy to follow. Figures are included and adequately labeled, enhancing the paper's clarity. However, the paper could benefit from some minor proofreading for grammar and typographical errors.

Experimental design

The paper presents original primary research within the scope of the journal. The research question is well-defined and meaningful, focusing on the classification of cardiac arrhythmias using IIR filters and machine learning/deep learning models. The study addresses an identified knowledge gap related to noise interference in ECG readings. The investigation is conducted rigorously to high technical and ethical standards.

The methods section provides sufficient detail to replicate the study, including the use of the MIT-BIH database, IIR filter design, and data preprocessing techniques. However, more details on the machine learning and deep learning models employed would be beneficial to readers. Specifically, the hyperparameters of the models should be described.

Validity of the findings

The paper does not assess the impact and novelty of the findings, which could provide valuable insights into the significance of the research. Encouraging meaningful replication by providing a clear rationale and benefit to the literature is a positive aspect of the paper. The provision of underlying data is not mentioned explicitly, but it is important for transparency and reproducibility in scientific research. The conclusions are generally well-stated and linked to the original research question. However, the paper could benefit from a more concise and focused discussion of the findings.

Additional comments

Overall, the paper presents an interesting and relevant study on arrhythmia classification using IIR filters and machine learning/deep learning models. It addresses a critical healthcare issue and proposes a novel approach. To enhance the paper's quality, consider the following suggestions:

Provide more details on the machine learning and deep learning models used, including hyperparameters.
Assess the impact and novelty of the findings to highlight the significance of the research.
Explicitly mention the provision of underlying data for transparency.
Proofread the paper for grammar and typographical errors.

---

## Round 0.2 · Minor Revisions

Several problems raised by the first reviewer are still present in the article.
The article cannot be recommended for acceptance right now; the authors should take care of the issues highlighted and prepare a new version of the manuscript.

·

Basic reporting

The authors assert that their study's primary objective is to develop a model incorporating both machine learning and deep learning approaches, complete with hyperparameter tuning—a method not previously explored in the literature, rendering it a unique contribution to the field. However, the reviewer contests this claim, pointing out instances where both traditional machine learning and deep learning models, coupled with hyperparameter optimization, have been applied successfully. For example, Ullah et al achieved a 99% accuracy in classifying MIT-BIH data using a CNN (PMID: 36210977).

Additionally, the authors contend that their proposed IIR filter for noise reduction in ECG signals is a novel approach without precedent in machine learning and deep learning research. However, the reviewer refutes this assertion, referencing the article titled "An Efficient Designing of IIR Filter for ECG Signal Classification Using MATLAB" by Manjula et al, published in 2023, which addresses a similar application.

In light of the above, it is apparent that the authors have not adequately addressed the reviewers' comments in these instances. As a clinical practitioner, the reviewer typically emphasizes clinical implications in their peer review comments but found no discernible novelty in the research presented in this manuscript.

Experimental design

See previous comments.

Validity of the findings

See previous comments.

Additional comments

See previous comments.

Reviewer 3 ·

Basic reporting

Some minor grammar and typographical errors have been addressed.

Experimental design

The details on the machine learning and deep learning models employed have been improved, specifically, the hyperparameters of the models.

Validity of the findings

The paper has addressed the novelty of the findings, which provide valuable insights into the significance of the research.

Additional comments

The authors have addressed my questions.

---

## Round 0.3 · accepted · Accept

The authors sufficiently address the issues raised by the reviewers and therefore I can recommend this article for acceptance.